# Quantifying the role of surface plasmon excitation and hot carrier transport in plasmonic devices

Giulia Tagliabue[1,2], Adam S. Jermyn[3], Ravishankar Sundararaman[4], Alex J. Welch[1,2], Joseph S. DuChene[1,2], Ragip Pala[1], Artur R. Davoyan[1,5,6], Prineha Narang[7] & Harry A. Atwater [1,2,6]

Harnessing photoexcited "hot" carriers in metallic nanostructures could define a new phase of non-equilibrium optoelectronics for photodetection and photocatalysis. Surface plasmons are considered pivotal for enabling efficient operation of hot carrier devices. Clarifying the fundamental role of plasmon excitation is therefore critical for exploiting their full potential. Here, we measure the internal quantum efficiency in photoexcited gold (Au)–gallium nitride (GaN) Schottky diodes to elucidate and quantify the distinct roles of surface plasmon excitation, hot carrier transport, and carrier injection in device performance. We show that plasmon excitation does not influence the electronic processes occurring within the hot carrier device. Instead, the metal band structure and carrier transport processes dictate the observed hot carrier photocurrent distribution. The excellent agreement with parameter-free calculations indicates that photoexcited electrons generated in ultra-thin Au nanostructures impinge ballistically on the Au–GaN interface, suggesting the possibility for hot carrier collection without substantial energy losses via thermalization.

[1] Thomas J. Watson Laboratories of Applied Physics, California Institute of Technology, 1200 East California Boulevard, Pasadena, CA 91125, USA. [2] Joint Center for Artificial Photosynthesis, California Institute of Technology, 1200 East California Boulevard, Pasadena, CA 91125, USA. [3] Institute of Astronomy, Cambridge University, Cambridge CB3 0HA, UK. [4] Department of Materials Science and Engineering, Rensselaer Polytechnic Institute, 110 8th Street, Troy, NY 12180, USA. [5] Resnick Sustainability Institute, California Institute of Technology, Pasadena, CA 91125, USA. [6] Kavli Nanoscience Institute, California Institute of Technology, Pasadena, CA 91125, USA. [7] John A. Paulson School of Engineering and Applied Sciences, Harvard University, Cambridge, MA 02138, USA. Correspondence and requests for materials should be addressed to H.A.A. (email: haa@caltech.edu)

Efficient collection of photoexcited, non-equilibrium "hot" carriers within metallic nanostructures offers considerable promise for band gap-free photodetection and selective photocatalysis[1,2]. However, practical applications require significant improvements in the performance of hot carrier devices relative to current performance. Excitation of surface plasmon polaritons—hybrid light-matter states localized at a metallic interface—is commonly viewed as a promising pathway for boosting the efficiency of these systems[1–13]. Indeed, numerous experimental studies based on internal photoemission[14–16] (IPE) of hot electrons in metal–semiconductor photodiodes have shown a close correlation between the plasmonic resonance of the nanoantenna and the device responsivity (i.e., light-to-current conversion)[12,17–23]. Such close correlation suggested that the dramatically enhanced optical near-fields associated with surface plasmon excitation may alter the quantum efficiency of hot carrier generation and collection[7,8,13,18]. To date, however, a detailed analysis distinguishing the role of plasmon excitation from hot carrier transport and injection in these systems has remained elusive.

The responsivity of a photodetector, or equivalently its external quantum efficiency (EQE), describes the overall efficiency with which the device converts incident photons to collected electrons (Fig. 1a). However, this metric convolutes the effects of plasmonic absorption with the subsequent electronic relaxation and transport processes that occur within the device. While surface plasmons are well known to enhance light absorption[24] (Fig. 1b), deeper insight into their fundamental role in the physics of hot carrier devices requires a careful analysis of the internal quantum efficiency (IQE, Fig. 1c), which deconvolutes absorption and transport. Indeed, IQE is an established measure for evaluating interband processes in semiconductor optoelectronics[25]. Yet, experimental studies of IQE in plasmonic hot carrier IPE systems to date have provided limited understanding of plasmon-mediated hot carrier transport and injection. In particular, previous work has relied on a semi-classical Fowler theory for interpreting the experimental IQE spectra[17,21–23,26–29]. Failures of this approximation in the visible regime[18,23,30], where interband absorption in metals may be dominant, have required making ad hoc assumptions regarding the effect of plasmon excitation in electronic transport processes[26,31], in contrast with results of recent ab initio calculations[32]. Furthermore, a deeper experimental analysis of plasmonic hot carrier transport has so far been obscured by parasitic optical losses present in the plasmonic structures (e.g., from use of adhesion layers and by parasitic hot carrier relaxation and absorption away from the junction in nanostructures thicker than the hot carrier mean free path). Overall, a lack of systematic experimental measurements together with limited model fidelity have prevented a clear assessment of the physics underlying plasmon-derived hot carrier transport and collection.

In this work, we perform an experimental study to elucidate and quantify the role of plasmons in hot carrier devices. We assess the IQE of several hot carrier devices with distinct plasmonic resonances, which were designed to minimize parasitic effects, including optical loss and carrier relaxation. Our studies indicate that transport—as characterized by the IQE—is a distinct and independent process from carrier generation by plasmon excitation. With direct measurements, we deterministically conclude that plasmons solely affect the optical properties of the device without modifying the internal processes associated with hot carrier transport and collection. We also show that the metal electronic band structure and the metal–semiconductor interface influence device performance, particularly at photon energies above the interband absorption threshold. We further provide insight into hot carrier generation, transport, and collection in plasmonic-metal/semiconductor Schottky junctions by coupling spectrally resolved measurements of hot electron collection across Au/n-GaN heterojunctions with a recently developed parameter-free hot carrier transport model[33]. Going beyond a description of individual electronic processes[32,34–37], this combination of theory and experiment enables an accurate depiction of the complex interplay between hot carrier generation and transport in realistic experimental structures without ad hoc assumptions. In particular, our analysis reveals that the measured photocurrents arise from ballistically injected hot electrons at photon energies below the threshold for interband transitions (~2 eV).

## Results

**Experimental evaluation of IQE and the role of plasmon excitation.** To experimentally assess the role of plasmon excitation on hot electron device performance, it is necessary to decouple optical excitation from subsequent electronic transport and collection. For this purpose we experimentally compared several Au/GaN photodetector devices with distinct plasmon resonances but identical metal–semiconductor Schottky junctions. An abrupt plasmonic metal/semiconductor interface and plasmonic nanoantennas with thickness smaller than the hot carrier mean free path ($l_{mfp}$) are necessary to ensure maximal sensitivity to ballistically harvested carriers. Accordingly, our experimental platform consists of planar Au plasmonic photodiodes on an optically transparent yet highly electrically conductive n-type GaN substrate, that we have identified as an optimal support (see Methods) to enable coupled electrical and optical (both transmission and reflection) characterization throughout the entire ultraviolet/visible/near infrared spectral range. Each heterostructure consists of a large Au contact pad connected to an array of electrically conductive Au stripes, which serve as nanoantennas that support plasmon resonances in the Vis-NIR regime. For a fixed period ($P$) of 230 nm, specifically chosen to suppress diffraction orders in the wavelength ($\lambda$) range of interest, the spectral position of the dipolar plasmon mode is controlled by adjusting the stripe width ($W$). Three hot carrier heterostructures were constructed with $W$ of 61, 70, and 85 nm to achieve plasmon resonances located at ca. 1.9, 1.85, and 1.72 eV. The Au nanoantenna thickness ($t_{Au} = 20$ nm) approaches the expected average mean free path for hot carriers (ca. 10–20 nm at 2 eV[32]) and was chosen to maximize the collection of ballistic hot electrons without sacrificing optical absorption. A titanium (Ti) Ohmic contact completes the planar plasmonic diode so that photocurrent can be collected while illuminating the sample through the transparent sapphire substrate (Fig. 2a and Methods).

The formation of a Schottky barrier ($\Phi_B$ ~1.2 eV[38]) at the Au/n-GaN interface ensures that electron-hole pair separation occurs even in the absence of an external bias. As expected, we observed a linear relationship between the short-circuit photocurrent, $I_{sc}$ and incident laser power (Fig. 2b) when using a 633 nm diode laser to irradiate a stripe array ($W = 61$ nm). We attribute the linear photoresponse to the injection of hot electrons from the Au nanoantennas into the n-GaN conduction band, since the incident photon energy is much less that the bandgap of the semiconductor ($E_g = 3.4$ eV ~364 nm[39]). Furthermore, the large barrier for hot hole injection from the metal into the semiconductor valence band ($\Phi_{B,Hole} > 3$ eV) allows us to exclude any potential contribution from hot holes to the device photocurrent in the studied photon energy range.

For each heterostructure, steady-state EQE and absorption spectra are determined experimentally by measuring both the wavelength-dependent photocurrent as well as transmission and reflection spectra under the same illumination conditions of tunable, monochromatic light polarized perpendicular to the

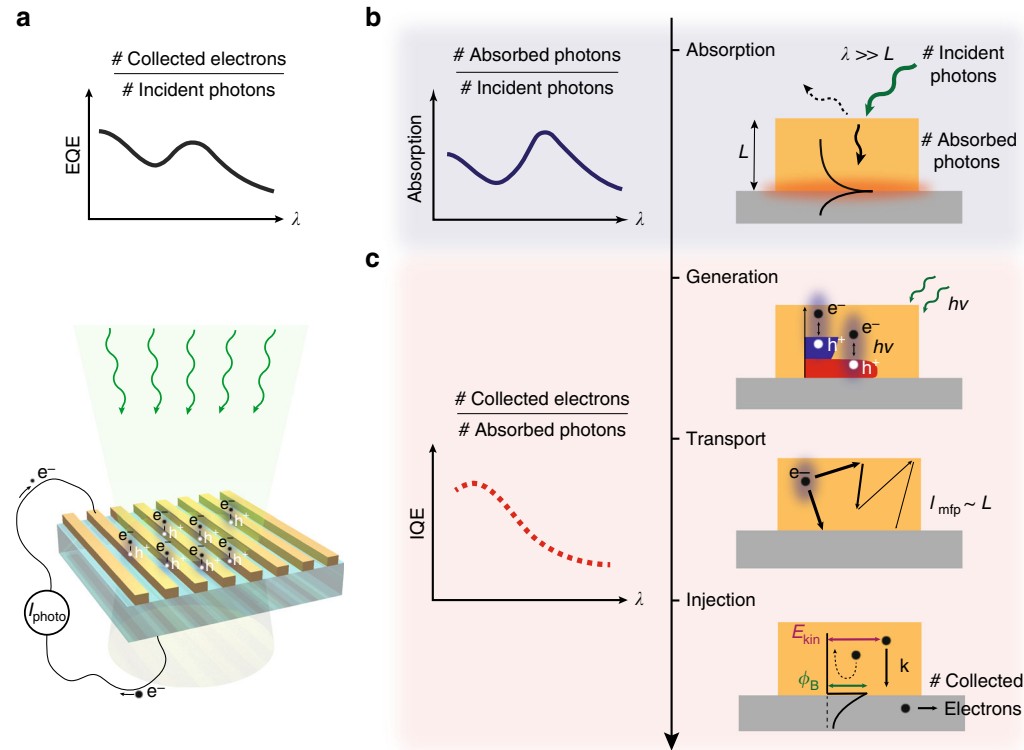

**Fig. 1** Carrier generation and transport in photoexcited metal nanostructures. **a** Schematic representation of carrier generation and transport via internal photoemission (IPE) in a plasmonic metal–semiconductor heterostructure: charge carriers created in the metal upon illumination are separated across the metal–semiconductor interface generating a photocurrent at sub-bandgap photon energies. The external quantum efficiency (EQE) spectrum represents the wavelength ($\lambda$)-dependent photon-to-electron conversion probability. As show in **b** and **c**, the EQE can be decomposed into the product of absorption and internal quantum efficiency (IQE); **b** Illustrative absorption spectrum of a metal nanostructure displaying a resonant plasmonic feature which can be engineered through photonic design. Plasmon excitation indeed yields high absorption in metallic nanostructures with characteristic dimension $L$ much smaller than the wavelength $\lambda$ of the incident photon; **c** Illustrative IQE spectrum and schematic representation of the electronic processes which contribute to it, i.e., generation of carriers through intraband and interband transitions, propagation, and scattering of the hot carriers with energy-dependent mean free path ($l_{mfp}$), and injection of hot carriers with adequate kinetic energy ($E_{kin}$) and momentum ($k$) across the Schottky barrier, $\Phi_B$

stripes (see Methods). For the heterostructure with $W = 61$ nm, a resonance peak at $\lambda_{peak} = 650$ nm can be observed in both spectra (Fig. 2c, e), absorption being in excellent agreement with numerical simulations (Fig. 2e, dashed line). Spatial maps of absorption in the photoelectrode were collected off-resonance above the interband threshold of Au ($\lambda = 514$ nm $< \lambda_{IB} \sim 688$ nm) as well as on-resonance ($\lambda_{peak} = 650$ nm). In the first case, the unpatterned Au pad exhibits larger absorption than the array of nanoantennas (Fig. 2d, $\lambda = 514$ nm). Instead, on resonance (Fig. 2d, $\lambda = 650$ nm), absorption in the plasmonic stripe array ($\approx 60\%$) greatly exceeds that of the Au film. It is noted that this feature disappears upon rotating the incident light polarization by 90° (Supplementary Notes 1 and 2). Such behavior confirms that the photocurrent originates from optical excitation of the dipolar plasmon mode in the nanoantennas. It is interesting to note that not only the plasmon resonance, but also the fringes present in the absorption spectrum (Fig. 2e), which are due to Fabry–Perot interference[40] in the planar GaN/sapphire substrate structure (Supplementary Note 7), cause a modulation in the photocurrent response that is reproduced in the EQE spectrum (Fig. 2c).

Comparing the optical (absorption) and electrical (EQE) performance of three hot carrier heterostructures with varying stripe width, we find a close correlation between the plasmon excitation wavelength and the EQE peak response (Fig. 2f). Increasing $W$ from 61 to 85 nm red shifts both absorption and EQE peak positions ($\lambda_{peak}$) to a commensurate amount (Supplementary Note 2). In contrast, the IQE spectra, determined by taking the ratio of EQE and absorption (Fig. 2g), do not exhibit

any spectral features that are associated with the characteristic peak wavelength for plasmonic absorption in each device (see Supplementary Note 7 regarding the residual Fabry–Perot fringes). The striking similarity of the three IQE curves indicates that the carrier transport and collection processes are the same in all three devices, even though the absorption spectra are different, suggesting that the role of plasmon excitation is primarily associated with optical absorption and not transport. That is, tunable plasmon resonances efficiently couple far-field radiation into nanoscale volumes and this mechanism dominates the EQE across a range of wavelengths. This observation implies that the intrinsic material properties of the metal and the interface barrier height dictate the transport characteristics of the heterostructure. Thus, plasmon excitation does not a priori selectively enhance the rate of any particular decay process or transport mechanism. Interestingly, as also remarked in previous studies[18,23,26], we observed that all three IQE curves were characterized by a broad, asymmetric feature peaking around 560–565 nm (~2.2 eV), which cannot be described by conventional Fowler models for IPE. Contrary to previous speculations about the role of indirect bandgap materials[18], our results on a direct bandgap semiconductor (n-GaN) indicate that it is the electronic band structure of the metal that determines the energy dependence of the IQE.

**Ab initio modeling of electronic processes and IQE.** For hot carriers, IQE is comprised of three distinct processes[25] (Fig. 1c): (i) generation of a non-equilibrium distribution of "hot" electrons

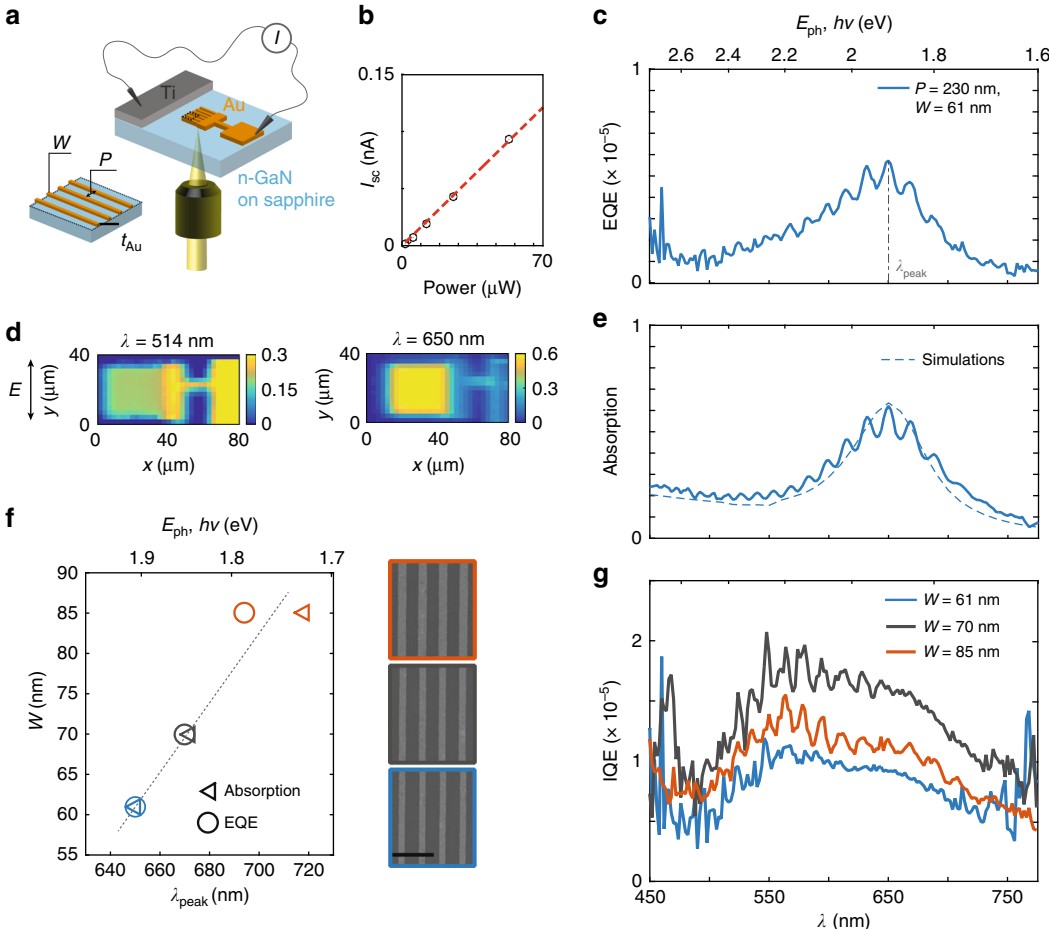

**Fig. 2** Role of plasmon excitation on hot electron IPE in metal–semiconductor heterostructures. **a** Schematic representation of the designed plasmonic heterostructures as well as measurement configuration: a 20 nm thick, nano-patterned gold (Au) photoelectrode is fabricated on n-type GaN (3.4 eV band gap, Schottky barrier $\Phi_B$ ~1.2 eV) together with a 75 nm thick titanium (Ti) Ohmic contact; light is incident on the plasmonic resonant Au nanostripe array (stripe width $W$, array period $P$ from the bottom and the photocurrent is collected via two microcontact probes); **b** short-circuit photocurrent $I_{sc}$ (i.e., 0 V applied bias) upon illumination of one heterostructure ($W = 61$ nm) with a diode laser ($\lambda_{laser} = 633$ nm) as a function of incident power; **c** EQE spectrum of the fabricated heterostructure with stripe width $W = 61$ nm and periodicity $P = 230$ nm exhibiting a resonance peak at $\lambda_{peak} = 650$ nm; **d** spatial maps of absorption for illumination of the Au photoelectrode off-resonance (514 nm–2.14 eV) and on-resonance (650 nm–1.9 eV) with light polarized perpendicular to the stripes; **e** measured (solid line) and simulated (dashed line) absorption spectra for the same heterostructure exhibiting a plasmon resonance at $\lambda_{peak} = 650$ nm; **f** EQE and absorption resonance peak wavelengths ($\lambda_{peak}$) for three heterostructures with constant array periodicity ($P = 230$ nm) and increasing nanostripe width, $W$, equal to 61 nm (blue), 70 nm (gray), and 85 nm (red), respectively. Representative SEM micrographs are shown on the right (scale bar = 500 nm); **g** IQE spectra of the three plasmonic heterostructures shown in part (**f**)

and holes in the metal nanostructure upon plasmon decay via intraband (sp-sp) and interband (d-sp) optical transitions[32]; (ii) transport of these hot carriers to an interface either ballistically or via electron–electron and electron–phonon scattering and relaxation[32]; (iii) injection of carriers with appropriate momenta and sufficient kinetic energy above the interfacial Schottky barrier ($\Phi_B$)[25]. We can relate the specific shape of the IQE curves to the interplay between the two hot carrier generation mechanisms, namely, intraband and interband transitions, as well as their corresponding hot carrier distributions relative to the Schottky barrier height present at the metal–semiconductor interface. The interband and intraband decay rates are determined from density functional theory (DFT) calculations, which generate the prompt hot electron energy distribution. For antennas with sizes of the order of tens of nanometers as in our study, quantization effects of the electronic levels of the metal can be neglected and the bulk properties of gold can be used. Devices employing metallic nanocrystals with dimensions smaller than a couple of nanometers would need to take this aspect into account[36]. The decay

rate is dependent on both incident photon energy and the electronic band structure of the metal[32,35]. For photon energies below the interband threshold of Au ($h\nu_{IB}$ ~1.8 eV), hot electrons generated via intraband transitions have a nearly uniform probability at all energies from the Fermi level up to the photon energy (Fig. 3a, solid red curve). As a result, intraband excitation accounts for a sizable fraction of the hot electron distribution at energies above the Schottky barrier height (gray shaded area in Fig. 3a). In this low photon energy regime there is very good agreement between the Fowler model, based on the parabolic band approximation, and full DFT calculations (compare solid red curve with dashed red curve in Fig. 3a). On the other hand, above $h\nu_{IB}$ (Fig. 3b, solid turquoise curve), a much higher probability distribution is observed for low-energy carriers, since hot electrons originate from d-band levels deep below the Au Fermi level[41]. Consequently, there is a substantial reduction in the fraction of high-energy electrons created from intraband transitions compared to that predicted for the case of a purely parabolic band (Fig. 3b, dashed turquoise curve). This interplay, combined

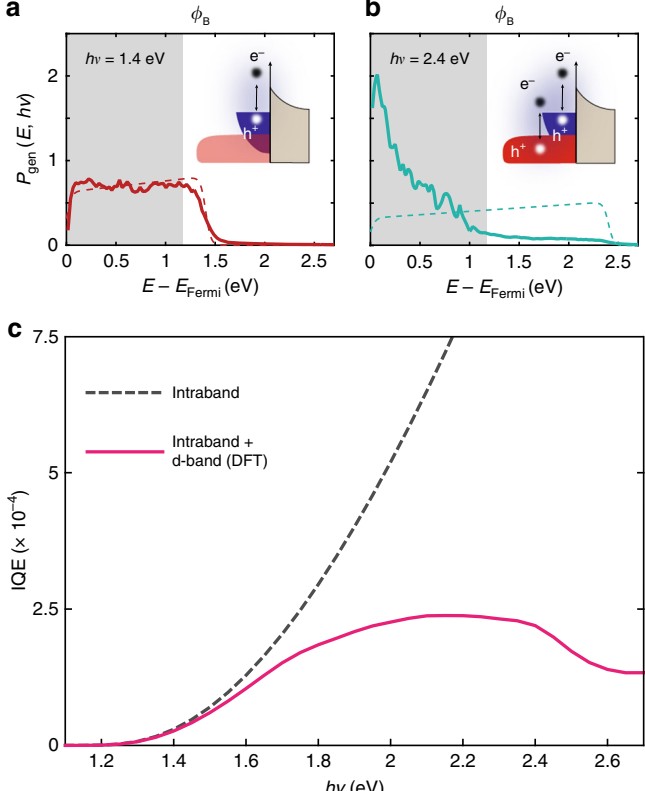

**Fig. 3** Impact of interband and intraband transitions on IQE of hot carrier devices. **a** Prompt hot electron energy distribution ($P_{gen}$) showing the carrier energy $E$ above the Au Fermi level ($E_F$) calculated with DFT (solid line) as well as under the parabolic band approximation (Fowler-like model, dashed line) for incident photon energies ($h\nu$) of 1.4 eV and **b** 2.4 eV. The shaded area in both plots depicts the position of the Schottky barrier, $\Phi_B$, limiting the possibility of collection to those carriers with energy $E-E_F > \Phi_B$. The insets show a schematic of the metal and semiconductor band structure illustrating the predominance of intraband transitions (**a**) and co-existence with interband transitions (**b**) as well as the presence of the Schottky barrier at the interface; **c** IQE spectra calculated based on the $P_{gen}$ obtained with DFT (magenta solid curve), i.e., including interband transitions, as well as with parabolic band approximation (gray dashed curve), i.e., accounting only for intraband transitions. For the injection process, conservation of tangential momentum is assumed[21]. Transport of hot electrons within the metal nanostructure has been neglected

with the height of our Schottky barrier (gray shaded area in Fig. 3a, b—see also Supplementary Note 3), results in a reduction in IQE at energies above the interband threshold (Fig. 3c, magenta solid line). This is in sharp contrast to predictions of the Fowler model, which accounts exclusively for intraband processes (Fig. 3c, gray dashed line). However, it must be recognized that even above $h\nu_{IB}$ both types of transitions occur simultaneously and high-energy carriers continue to be generated, though with decreasing probability. Plasmon excitation does not alter this interplay, as it does not directly influence the hot carrier distribution, only the number of photons absorbed by plasmon generation at a given frequency. Changes in the dominant optical transition mechanism with increasing photon energy explains why the metal band structure, and in particular its interband threshold, has such a profound effect on the overall IQE of hot carrier devices. Interestingly, recent DFT calculations[32] show that in the case of Al nanoantennas, interband transitions produce a hot electron/hot hole distribution which is very similar to the

intraband case and therefore IQE could preserve the quadratic dependence on the photon energy even above the interband threshold (~1.6 eV).

A microscopic understanding of hot carrier transport in Au/n-GaN heterostructures is obtained by comparing experimental measurements to results of a recently developed theoretical framework that combines electromagnetic simulations, ab initio DFT calculations, and Boltzmann transport methods to compute the generation and transport of hot carriers within realistically scaled (ca. 10–100 nm) metallic structures[33] (see Methods). From electromagnetic simulations, we first determine the electric field profile in a single Au nanoantenna ($W = 61$ nm, Fig. 4a and Supplementary Note 8). The initial energy and momentum distribution of the hot carriers are obtained from plasmon decay rates and electronic optical excitations derived from DFT calculations[32,35], which account for the anisotropies associated with these quantities in the interband regime as well as resistive contributions in the intraband regime. Energy-dependent life-times and mean free paths ($l_{mfp}$) are also calculated with ab initio methods accounting for both electron–electron and electron–phonon scattering processes and have been shown previously to agree very well with experimental results[34,42]. All the calculated quantities are averaged over different crystalline orientations to reflect the polycrystalline nature of the fabricated structures. This information is combined in a Boltzmann transport calculation[33] where we compute the propagation of carriers across the Au nanostructure, determining changes to their energy distribution as well as the number of scattering events they experience. For each photon energy, our calculations yield the energy-resolved flux $F_N(E)$ of hot electrons with energy $E$ above the metal Fermi level that reach the Au/n-GaN interface after up to $N$ scattering events. Attesting to the validity of our computational approach, the energy-resolved flux of hot electrons that reach the interface ballistically, $F_0(E)$, shown in Fig. 4b retains the key features described in Fig. 3a. The model also shows that scattering processes serve to homogenize the hot carrier distributions by smoothing the transition between the intraband and interband generated carriers that reach the interface (Fig. 4c).

Estimating the injection probability, $P_{inj}(E)$ across the Schottky barrier based on the assumption of tangential momentum conservation (Supplementary Note 4)[21], we then calculate IQE as $\Phi_B$:

$$\text{IQE} = \int_{\Phi_B}^{\infty} F_N(E) \cdot P_{inj}(E)\,dE \text{ for N} = 0, 1, \ldots$$

The blue solid curve in Fig. 4d represents the IQE spectrum predicted from $F_0(E)$ and the blue dashed curve is the predicted IQE obtained for $F_3(E)$. Including additional scattering events only changes the IQE by 0.01%, indicating that the vast majority of hot electrons undergo no more than three scattering events before being collected. Significantly, our parameter-free model of hot carrier generation, transport, and injection is in excellent quantitative agreement with the experimental data (gray solid curve).

## Discussion

This result shows that a detailed description of material properties and device geometry can precisely capture the details of plasmonic hot carrier transport under illumination, both on and off resonance. Strikingly, the results of our model indicate that more than 90% of the hot carriers are collected ballistically at photon energies below 2 eV ($\lambda > 620$ nm), implying that hot carrier transport in our Au nanoantennas occurs in the ballistic regime at the plasmon peak position. This result retrospectively

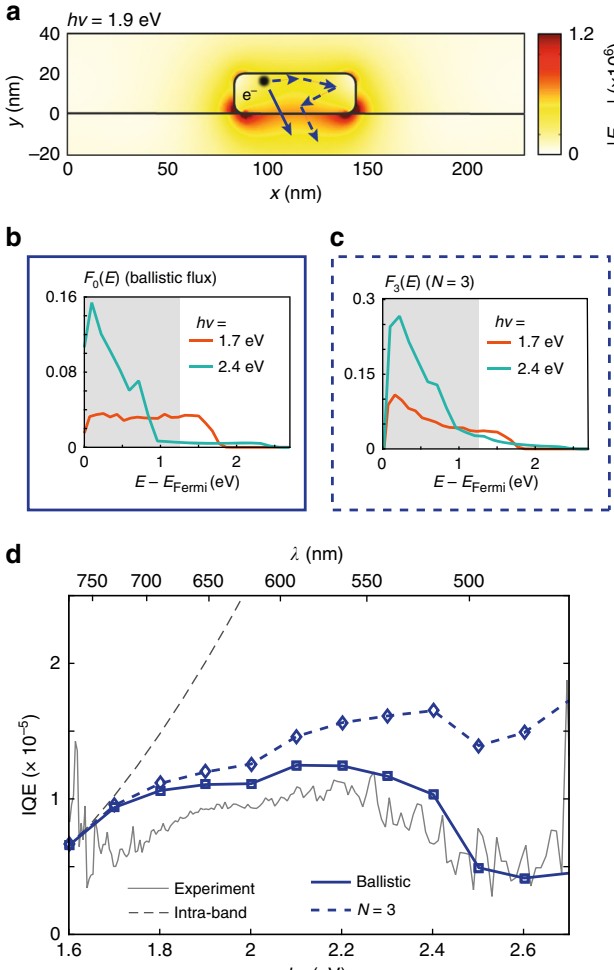

**Fig. 4** Hot electron generation and transport in plasmonic nanoantennas. **a** Calculated spatial profile of the electric field norm $|E_{field}|$ at resonance ($\lambda = 650$ nm, $h\nu = 1.9$ eV, $E_0 = 2.56 \times 10^5$ V m$^{-1}$) for the experimental structure with $W = 61$ nm and $P = 230$ nm. $|E_{field}|$ in the metal defines the spatial generation profile of the hot carriers. As schematically illustrated, hot electrons then propagate across the metal structure and reach the Au–GaN interface either ballistically (solid arrow) or after scattering (dashed arrows); **b** energy-resolved flux of hot electrons reaching the Au–GaN interface ballistically for photon energy of 1.7 eV (orange curves, weak interband contribution) and 2.4 eV (turquoise curves, strong interband contribution). The shaded area shows the position of the Schottky barrier; **c** same as **b** but including the flux of carriers that have undergone up to $N = 3$ scattering events; **d** IQE spectra calculated based on the computed energy-resolved fluxes, both for the ballistic case (blue solid curve) and for $N = 3$ (blue dashed curve), under the assumption of tangential momentum conservation for the injection probability[21]. The gray dashed curve represents the IQE estimated based on the fit of Fowler yield, $IQE_{Fowler} = C \cdot (h\nu - \Phi_B)^2 / h\nu$ with $\Phi_B$ ~1.2 eV and $C = 6.7 \times 10^{-5}$. The gray solid curve is the experimentally determined IQE (Fig. 2g, blue curve)

validates the tailored design of our experimental platform toward ballistic hot carrier collection. Increasing the thickness of the plasmonic antenna would increase the contribution of scattered carriers to the observed photocurrent, due to the increased distance that hot electrons must travel before reaching the interface. Since each electron–electron scattering event approximately reduces the electron energy by a factor of two, it is expected that scattered carriers would only provide a significant contribution at higher photon energies; those carriers created at lower photon

energies would likely have insufficient energy to overcome the Schottky barrier. Nonetheless, in the studied configuration, which is very common for plasmonic photodetectors, the plasmonic antenna sits on a high-index GaN substrate, and therefore the electric field is localized close to the metal–semiconductor interface upon excitation of the fundamental plasmon mode (Fig. 4a). The largest hot carrier generation thus occurs close to the interface, and as a result, the non-uniform field profile inside the antenna favors ballistic collection, mitigating the effect of increasing antenna thickness. Therefore, by enabling strong light localization in metallic nanostructures (Fig. 4a) plasmon excitation may be able to realize optoelectronic systems that operate in the truly ballistic regime despite the short, energy-dependent $l_{mfp}$ of hot electrons in metals.

We also observe that our experimental IQE values agree quantitatively with theoretical results based on metal electronic structure, suggesting that the collection efficiency is limited by fundamental electronic structures characteristics of the metal and interface. To summarize, the key aspects influencing IQE are: (i) the metal band structure, (ii) the transport processes to the interface, (iii) the Schottky barrier height, and (iv) the momentum matching condition for injection across the interface. We note that the momentum matching factor has profound consequences for the overall magnitude of the IQE. Indeed, the low effective electron mass in GaN[43] and the smooth metal–semiconductor interface in our devices, which imposes tangential momentum conservation, account for a reduction in IQE by nearly four orders of magnitude (Supplementary Note 5). Use of semiconductors with heavy electrons or large density of states in the conduction band (e.g., TiO$_2$) as well as nanoscopically roughened metal–semiconductor interfaces could thus be beneficial to boost the IQE and performance of hot carriers IPE devices. Irrespective of the Schottky barrier height, momentum matching conditions also cause a disproportionate suppression in the collection of low-energy electrons originating from either interband transitions or scattering of high-energy carriers generated by intraband transitions. Therefore the metal–semiconductor interface plays a significant role in the ultimate efficiency of plasmonic hot carrier IPE devices. We also note here that plasmon-mediated interfacial hot carrier excitation has been observed in selected systems employing small metallic nanocrystals and constitutes a different mechanism for harnessing hot carriers beyond IPE[44,45]. In fact, in the case of interfacial plasmon excitation the quantum efficiency has been shown to exhibit a stepwise efficiency spectrum with a system-specific threshold energy[44]. However, in the studied systems, which have dimensions of several tens of nanometers, we can entirely ascribe the IQE spectral features to the metal properties and we do not observe any deviations that could be attributed to a competing contribution from interfacial plasmon excitation. Transport of carriers from their point of generation to the interface, where they are filtered by the presence of a sizeable Schottky barrier ($\Phi_B$ ~1.2 eV), accounts for the remaining one to two orders of magnitude reduction in IQE. It is worth noting that even assuming a 50 meV Schottky barrier, values of IQE ~10$^{-4}$ are expected for this system (Supplementary Note 5). Considering these factors, we suggest that a potentially promising strategy for increasing the IQE value is to identify metals with a high density of states close to the Fermi level, which would enable the efficient creation of hot electrons with high energies and offer an interesting path toward high-performance hot carrier devices. Simultaneously, careful design of the device geometry[19,30,46] and further engineering of the spatial hot carrier generation profile could promote ballistic collection, and hence improve device efficiency.

To summarize, our experimental analysis of IQE in ultrathin plasmonic nanoantennas with abrupt metal/semiconductor

interfaces reveals that plasmon excitation enables the efficient coupling of far-field radiation into nanoscale volumes, but does not dictate the transport physics governing the performance of hot carrier photoemission devices. Instead, analysis of the IQE spectra emphasizes the role of interband and intraband decay processes, as well as carrier transport over nanometer scale distance in the metal, in determining the distribution of hot carriers that are collected via IPE. Our observation of ballistic electrons is encouraging for efforts to use ballistic hot carrier collection for ultrafast photodetection and excited-state photocatalysis. Our results reveal mechanisms important to the design of efficient hot carrier devices, and they suggest that new materials with tailored band structure and transport properties will be crucial for the realization of efficient hot carrier-driven devices. Future experiments using ultrafast spectroscopy techniques and time-resolved IQE measurements may expand our understanding of hot carrier transport, and allow for more comprehensive comparison with theoretical predictions. As an outlook, the agreement between our experimental data and detailed, parameter-free theoretical hot carrier transport model suggests that this combined approach can be a powerful tool to guide the design of future hot carrier optoelectronic devices.

## Methods

**Sample fabrication.** In order to perform coupled optical and electrical measurements of a plasmonic IPE device for experimental assessment of its IQE, it is necessary to have a semiconducting substrate which: (i) does not absorb light in the wavelength range of interest in order to prevent interband photogeneration of carriers within the semiconductor and also does not scatter light (optically transparent); (ii) has high electrical conductivity to enable transport of hot carriers; and (iii) forms a Schottky barrier with the metal to favor separation of the hot electrons and holes and to prevent their recombination. The n-GaN substrate employed here satisfies all of these requirements: (i) it has a wide bandgap (3.4 eV), and is optically transparent, with no light scattering centers; (ii) due to its widespread use in optoelectronics, it is commercially available with various doping levels, in the form of highly doped low electrical resistance, crystalline substrates; (iii) its band alignment leads to the formation of a sizable Schottky barrier of ~1.2 eV with Au. These factors motivated our use of GaN as a semiconducting support for the study of plasmonic IPE devices. GaN films on sapphire were purchased from Xiamen ($4 \pm 1\,\mu$m thick GaN layer, Ga-face, epi-ready, $N_d = 5–7 \times 10^{17}\,cm^{-3}$, $\rho < 0.5\,\Omega$ cm, $<10^8$ dislocations/cm²). A layer of S1813 was spin coated on the substrate (40 s, 3000 rpm) and post-baked for 2 min at 115 °C. The Ohmic pattern was exposed for 40 s and then developed for 10 s in MF319®. Then, 75 nm of Ti were deposited with e-beam evaporation (1.5 Å/s, base pressure lower than $5 \times 10^{-7}$ Torr). A layer of PMMA 495-A4 was spin coated on the sample (1 min, 4000 rpm) and baked for 2 min at 180 °C. Next, a layer of PMMA 950-A2 was spin coated on top of it (1 min, 5000 rpm) and also baked for 2 min at 180 °C. Then, e-beam lithography was used to write the nanoantenna pattern (Quanta FEI, NPGS System). Beam currents of approximately 40 pA were used with exposures ranging from 350 to 500 µC/cm², thus achieving different stripe widths with equal pitch. A 20 nm Au layer was then deposited with e-beam evaporation (Lesker) (0.8 Å/s, base pressure lower than $2 \times 10^{-7}$ Torr). Importantly, before any metal deposition, the sample was exposed to a mild oxygen plasma (30 s, 200 W, 300 mT) to remove any photoresist residual, dipped in a 1:15 $NH_4OH$:DI $H_2O$ solution for 30 s to remove any surface oxide layer and finally rinsed in water (30 s) and blown dry with $N_2$ gas. The substrate was then immediately loaded into the e-beam evaporator chamber, minimizing the time of exposure to ambient atmosphere.

**Photocurrent measurements.** A Fianium laser (2 W) was used as the light source for plasmon excitation. The beam was monochromated (slit width 200 µm), collimated, and finally focused onto the sample with a long working distance, low NA objective (Mitutoyo 5×, NA = 0.14). A Si photodetector was used to measure the transmitted power or, using a beam splitter, the reflected power incident on the sample. A silver mirror (M, Thorlabs) was used to normalize the reflection and the background (BG) was subtracted from all the measurements. A tilted glass slide was used to deflect a small amount of incident power from the laser onto a reference photodiode for coincident recording of the laser power incident on the sample. A chopper, typically at a frequency of ~100 Hz, was used to modulate the incident power and thus the photocurrent signal, which was subsequently processed with a lock-in amplifier. An external, low-noise current-to-voltage amplifier was used to feed the signal to the lock-in. Piezoelectric micro-probes (Mibots®) are utilized to electrically contact the sample and perform all of the photocurrent measurements.

**Numerical simulations.** A commercial finite element method software (COMSOL) is used to perform the electromagnetic simulations. The 3D simulations are performed to estimate absolute absorption values as well as 3D internal electric field distributions to be used in the subsequent hot carrier generation and transport code. The scattered field formulation is utilized. For the background field calculation, a port boundary condition with excitation "ON" is used to launch a plane wave with normal incidence and variable wavelength as well as for the recording of the reflected wave. A second port boundary condition without excitation is used to record the transmitted wave. Perfect magnetic conductor and periodic boundary conditions are used on the side walls (width of the unit cell equal to the array pitch, $P = 230$ nm, length of the cell equal to 50 nm). For the calculation of the scattered field, perfect-matched layers are used in place of the port boundary conditions.

**Hot carrier generation and transport predictions.** The hot carrier flux is computed by iteratively evaluating the effects of transport and scattering. In each iteration, transport effects are computed using the 1D Green's function ($\exp(-x/l_{mfp})$), where $l_{mfp}$ is the mean free path) on a tetrahedral mesh. Multiple different directions are integrated via Monte Carlo sampling. This results in a deposition of transported carriers at the surface and scattered carriers in the interior. The scattered carriers are then transformed via the scattering matrix elements to produce a new energy distribution at each point in the mesh, which is used as the input to the next round of transport calculations. The initial input distribution is obtained using the carrier energy-resolved dielectric function $\mathrm{Im}\varepsilon(\omega, E)$ and the input electromagnetic field from COMSOL, evaluated on the same tetrahedral mesh. $\mathrm{Im}\varepsilon(\omega, E)$ and the energy-dependent mean free path $l_{mfp}(E)$ are obtained using Fermi's golden rule, with electron–phonon and electron–photon matrix elements calculated using the DFT software JDFTx[47] (see ref. [33] for further details).

**Code availability.** First principle methodologies available through open-source software, JDFTx, and post-processing scripts available from authors upon request.

**Data availability.** All relevant data are available from the authors upon request.

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

## Acknowledgements

This material is based on work performed by the Joint Center for Artificial Photosynthesis, a DOE Energy Innovation Hub, supported through the Office of Science of the U.S. Department of Energy under Award No. DE-SC0004993. R.S., A.S.J., and P.N. acknowledge support from NG NEXT at Northrop Grumman Corporation. Calculations in this work used the National Energy Research Scientific Computing Center, a DOE Office of Science User Facility supported by the Office of Science of the U.S. Department of Energy under Contract No. DE-AC02–05CH11231. A.D. and H.A.A. acknowledge support from the Air Force Office of Scientific Research under grant FA9550-16-1-0019. G.T. acknowledges support from the Swiss National Science Foundation through the Early Postdoc Mobility Fellowship, grant no. P2EZP2_159101. P.N. acknowledges support from the Harvard University Center for the Environment (HUCE). A.S.J. thanks the UK Marshall Commission and the US Goldwater Scholarship for financial support. A.J.W. acknowledges support from the National Science Foundation (NSF) under Award No. 2016217021.

## Author contributions

G.T. performed experiments, numerical simulations, and IQE calculations of devices. A. S.J., R.S., and P.N. performed ab initio hot carrier generation and transport calculations. A.J.W., J.S.D., R.P., and A.R.D. contributed to experiments and data analysis. All authors contributed to interpretation of the results. G.T., J.S.D., A.R.D., and H.A.A. wrote the manuscript with contributions from all authors. H.A.A. supervised all aspects of the project.

## Additional information

**Competing interests:** The authors declare no competing interests.

