## [Peer Review File · Nature Communications]

Reviewers' comments:

Reviewer #1 (Remarks to the Author):

In the reviewed manuscript, the authors presents a detailed experimental and theoretical analysis of the hot electron generation, transport, and injection in a metal/semiconductor structure. In particular, the authors are able to quantify the efficiency of this process, separating the contribution of the internal quantum efficiency from that of the absorption of the metallic structure. I think the results are very timely and clearly appealing for a broad audience, and the claims well supported by the experimental and theoretical analysis. For these reasons, I recommend accepting the manuscript for publication. That said, I think the paper could benefit from the following suggestions:

- 1) It will be interesting if the authors could explain why the simulations of the absorption of the device shown in Figure 2e do not capture the oscillations
- 2) Are the DFT calculations shown in Figure 3 for bulk gold? If so, how do you expect the inclusion of the finite geometry should influence the results. The small thinckness of the structures could lead to a significant quantization of the electronic levels.
- 3) In the abstract the authors claim that "Despite numerous experiments, conclusive evidence that plasmon excitation imparts a fundamental advantage to the performance of hot carrier photoemission devices has not been shown." I think this sentence is misleading, and not necessary to support their claims. Indeed, in my view, many of the references the authors cite have shown evidence of the fundamental advantage of plasmon excitation on hot-carrier generation. There are, as well, other recent papers (see for instance [1,2]) that clearly demonstrate this.

[1] M. E. Sykes, et al. "Enhanced generation and anisotropic Coulomb scattering of hot electrons in an ultra-broadband plasmonic nanopatch metasurface", Nat. Commun. 8 986 (2017).

[2] A. E. Schlather, et al. "Hot Hole Photoelectrochemistry on Au@SiO₂@Au Nanoparticles", J. Phys. Chem. Lett. 8 2060 (2017).

Reviewer #2 (Remarks to the Author):

In their work, 'Quantifying the Role of Surface Plasmon Excitation and Hot Carrier Transport in Plasmonic Devices' G. Tagliabue and coworkers show, through experiments and theoretical calculations, how plasmon induced hot carrier generation in Au stripes on a GaN substrate is a decoupled process from hot carrier transport and transfer across an Au/ n-GaN Schottky barrier. Whereas the plasmonic resonance enhances the radiation absorption efficiency in the gold stripes (thus increasing the number of excited hot carriers), the metal's electronic structure, the transport mechanisms and the material properties of the substrate ultimately dictate the harvesting performance (i.e. current through the device) of the generated hot carriers.

In order to prove this, they extracted the Internal Quantum Efficiency (or IQE, which describes hot carrier dynamics upon carriers generation through the quantity: collected electrons vs absorbed photons) of the Au/n-GaN device from the External Quantum Efficiency (or EQE, which quantifies the overall process performance in terms of collected electrons vs. incident photons). While EQE spectra broadly follow the plasmonic absorption features (which are strongly dependent on the stripes geometry because of the different spectral positions of the plasmonic resonances), IQE spectra (where basically EQE is normalized by the device photon absorption) are found to be

almost geometry independent, thus suggesting that intrinsic bulk, interface characteristics and substrate properties mainly establish carrier dynamics and collection. A combination of FEM, DFT and Boltzmann transport calculations are capable to reproduce the experimentally obtained IQE spectra thus reinforcing the main claims of the work.

Additionally, the theoretical model provided insights about the physical mechanisms at the base of the electron collection in the device: besides elucidating the role of intraband and interband generated hot electrons, transport calculations predict that most of the collected carriers are ballistic thus furnishing guidelines for future device design based on ultra-thin metallic layers. The understanding of how hot carriers are generated and how they can be efficiently utilized is of broad significance. The topic treated in the manuscript is therefore of high interest not only in the plasmon, but, for instance, also in the optoelectronics or photocatalysis communities.

I can therefore suggest the publication in Nature Communications after minor revisions which should address the comments below:

1. I guess the experimental absorption spectra were possible thanks to transmissivity and reflectivity measurements through the transparent substrate. Can the author provide the spectra the T and R spectra (in the SI) for one or two geometry? I think it can be instructive for the reader to see the where the absorption features come from.
2. EQE and absorption fast oscillations (Fig. 2c and e) are probably related, as mentioned by the authors, to FP oscillations. Can the author find a relationship with the thickness of the substrate?
3. Fig 2b shows a linear correspondence between power and sc current. What the authors expect at higher powers and thus higher lattice temperatures? Would the temperature impact on the scattering frequencies modify the relative number of collected ballistic electrons?
4. How the thickness of the stripes would influence the mix of ballistic vs scattered electrons collection?
5. What would happen to IQE shape (fig 3c for instance) by using a metallic material (e.g. aluminum) where the interband transition is at lower energy than the plasmonic resonance (for properly designed stripes)?
6. Fig. 4c: the spike of the experimental (grey) at the right end of the plot is just an oscillation due to experimental instabilities or the IQE trend changes even further at higher energies?

Reviewer #3 (Remarks to the Author):

Major comment: The authors studied about the role of surface plasmon excitation in hot carrier generation, transport and injection by exploring internal quantum efficiency of photoexcited Au-GaN Schottky diodes. The authors concluded that the light-harvesting capability and strong near-fields of surface plasmon do not alter carrier generation rates or the internal device physics governing carrier transport, and that plasmon excitation does not a priori selectively enhance the rate of any particular decay process or transport mechanism. However, despite mentioning such important things, it is unclear why GaN which is not common for hot electron transfer studies was used. Especially, it is inappropriate to discuss in the same way as the past studies using Au-NPs loaded TiO₂ showing EQE from 0.1 to 1% by using the devices with very low EQE and IQE in the order of 10⁻⁵. IQE is the sum of all processes, and such discussion can only be done by measuring dynamics of each process such as dephasing of LSPR concerning with a hot carrier generation and excitation wavelength dependence of injection rate of hot electron into GaN electron conduction band. Furthermore, the near-field enhancement of Au line and space pattern is

very low because the electromagnetic field is not localized strongly (Even its near-field spectrum has not been measured.). The authors are needed to perform experiments by using hot carrier devices with high EQE efficiency and Au nanostructures with high near-field enhancement including more detailed measurements of not only IQE but also near-field spectrum as well as the dynamics of each process. Otherwise, it cannot be believed that IQE was not dependent on the localized surface plasmon and its near-field enhancement. Therefore, this version of manuscript cannot be accepted for publication in Nature Communications.

Minor comments: This manuscript does not show the I-t curve responding to the light turning on and off and its photocurrent stability. I-V curve measurement is also important. By using polarization dependence (difference in EQE action spectra between transverse and longitudinal modes), the photocurrent contributed from interband transition can be eliminated in the case of the Au line and space patterns or Au nanorods with aligned orientation. Namely, EQE spectrum only derived from surface plasmon can be obtained. However, the authors did not show the polarization dependence. The interference pattern was observed in the absorption spectrum because a GaN film with a thickness of about 4 micrometers is formed on a sapphire substrate. However, it is unclear why even the IQE spectrum also shows interference pattern. This indicates that the IQE spectrum was not obtained correctly.

Reviewer #1 (Remarks to the Author):

In the reviewed manuscript, the authors presents a detailed experimental and theoretical analysis of the hot electron generation, transport, and injection in a metal/semiconductor structure. In particular, the authors are able to quantify the efficiency of this process, separating the contribution of the internal quantum efficiency from that of the absorption of the metallic structure. I think the results are very timely and clearly appealing for a broad audience, and the claims well supported by the experimental and theoretical analysis. For these reasons, I recommend accepting the manuscript for publication. That said, I think the paper could benefit from the following suggestions:

Q1) It will be interesting if the authors could explain why the simulations of the absorption of the device shown in Figure 2e do not capture the oscillations

As the reviewer correctly observe, our calculated absorption spectra do not capture the experimental oscillations. Indeed, to reduce the computation time, in our numerical model we consider a GaN substrate that is thinner than the $\sim 4\mu\text{m}$ thick substrate we use in the experiments. Changes in the GaN thickness do not affect the plasmon peak position, however, a thinner GaN substrate does not support Fabry-Perot oscillations that, in the experimental case, end up modulating the absorption spectrum. The figure on the right shows the result of a simulation including the GaN substrate with a thickness close to the experimental value. The expected Fabry-Perot interference pattern can indeed be observed. The exact location of the fringes is very sensitive to the absolute GaN thickness, which

according to the manufacturer, varies by approximately $1\mu\text{m}$ across the wafer, and this variability causes uncertainties between the simulated and calculated spectra. Since these fringes are only related to the GaN thickness, they do not affect the hot carrier science we are reporting in this manuscript. Therefore, in order to preserve the clarity of the data in the manuscript we have added this figure in the new supplementary section S6, which is entirely devoted to the discussion of the Fabry-Perot interference modes.

Q2) Are the DFT calculations shown in Figure 3 for bulk gold? If so, how do you expect the inclusion of the finite geometry should influence the results. The small thickness of the structures could lead to a significant quantization of the electronic levels.

It is known (see for example Fundamentals of Microfabrication and Nanotechnology, Vol. I, Madou, CRC Press, 2011) that quantization of the electronic levels in a metallic system becomes important with level discreteness approaching the eV scale for 1-2 nm sized particles. The structures investigated here have dimensions of several tens of nanometers (20 -100 nm) along all three spatial directions and therefore the effect of the quantization of the electronic levels can be neglected. Using the bulk gold properties and accounting for the polycrystalline structure of the gold films is thus a valid approximation for this experimental system. Following the reviewer's suggestion, we have added the following sentence to the manuscript:

"For antennas with sizes of the order of tens of nanometers as in our study, quantization effects of the electronic levels of the metal can be neglected and the bulk properties of gold can be used. Devices employing metallic nanocrystals with dimensions smaller than a couple of nanometers would need to take this aspect into account³⁶."

Q3) In the abstract the authors claim that "Despite numerous experiments, conclusive evidence that plasmon excitation imparts a fundamental advantage to the performance of hot carrier photoemission devices has not been shown." I think this sentence is misleading, and not necessary to support their claims. Indeed, in my view, many of the references the authors cite have shown evidence of the fundamental advantage of plasmon excitation on hot carrier generation. There are, as well, other recent papers (see for instance [1,2]) that clearly demonstrate

this.

[1] M. E. Sykes, et al. "Enhanced generation and anisotropic Coulomb scattering of hot electrons in an ultra-broadband plasmonic nanopatch metasurface", *Nat. Commun.* 8 986 (2017).

[2] A. E. Schlather, et al. "Hot Hole Photoelectrochemistry on Au@SiO₂@Au Nanoparticles", *J. Phys. Chem. Lett.* 8 2060 (2017).

We thank the reviewer for pointing this out. We have corrected the abstract by replacing the misleading sentence with a more appropriate statement pointing to the importance of fundamental studies elucidating the role of plasmon excitation for the realization of non-equilibrium optoelectronic devices. The revised manuscript abstract now reads:

"Advancements in understanding of the fundamental role of plasmon excitation in hot carrier optoelectronics is therefore crucial to pave the way towards the full exploitation of their potential."

We have also included the suggested references in our revised manuscript, now listed as Ref. 11 and 13.

Reviewer #2 (Remarks to the Author):

In their work, 'Quantifying the Role of Surface Plasmon Excitation and Hot Carrier Transport in Plasmonic Devices' G. Tagliabue and coworkers show, through experiments and theoretical calculations, how plasmon induced hot carrier generation in Au stripes on a GaN substrate is a decoupled process from hot carrier transport and transfer across an Au/ n-GaN Schottky barrier. Whereas the plasmonic resonance enhances the radiation absorption efficiency in the gold stripes (thus increasing the number of excited hot carriers), the metal's electronic structure, the transport mechanisms and the material properties of the substrate ultimately dictate the harvesting performance (i.e. current through the device) of the generated hot carriers.

In order to prove this, they extracted the Internal Quantum Efficiency (or IQE, which describes hot carrier dynamics upon carriers generation through the quantity: collected electrons vs absorbed photons) of the Au/n-GaN device from the External Quantum Efficiency (or EQE, which quantifies the overall process performance in terms of collected electrons vs. incident photons). While EQE spectra broadly follow the plasmonic absorption features (which are strongly dependent on the stripes geometry because of the different spectral positions of the plasmonic resonances), IQE spectra (where basically EQE is normalized by the device photon absorption) are found to be almost geometry independent, thus suggesting that intrinsic bulk, interface characteristics and substrate properties mainly establish carrier dynamics and collection.

A combination of FEM, DFT and Boltzmann transport calculations are capable to reproduce the experimentally obtained IQE spectra thus reinforcing the main claims of the work.

Additionally, the theoretical model provided insights about the physical mechanisms at the base of the electron collection in the device: besides elucidating the role of intraband and interband generated hot electrons, transport calculations predict that most of the collected carriers are ballistic thus furnishing guidelines for future device design based on ultra-thin metallic layers.

The understanding of how hot carriers are generated and how they can be efficiently utilized is of broad significance. The topic treated in the manuscript is therefore of high interest not only in the plasmon, but, for instance, also in the optoelectronics or photocatalysis communities.

I can therefore suggest the publication in Nature Communications after minor revisions which should address the comments below:

1. I guess the experimental absorption spectra were possible thanks to transmissivity and reflectivity measurements through the transparent substrate. Can the author provide the spectra the T and R spectra (in the SI) for one or two geometry? I think it can be instructive for the reader to see the where the absorption features come from.

We thank the reviewer for this suggestion. We have revised Figure S2 by including the transmission and reflection spectra (measured with light polarization perpendicular and parallel to the stripes) of the device with stripe width $W = 61$ nm, which is the device shown in Figure 2 of the main manuscript. It is indeed instructive to observe the pronounced effect of the plasmon resonance in the transmission spectrum compared to the reflection spectrum.

2. EQE and absorption fast oscillations (Fig. 2c and e) are probably related, as mentioned by the authors, to FP oscillations. Can the author find a relationship with the thickness of the substrate?

We are happy to provide the reviewer the requested model for Fabry-Perot oscillations. In a Fabry-Perot etalon the peaks in transmission correspond to dips in reflection and vice versa. The peaks occur when the phase accumulation of the light wave during a round-trip across the high-index region (GaN in our case) is equal to 2π (constructive interference). If the refractive index and thickness of the GaN layer are n and t , respectively, we can calculate the phase accumulation as: $\delta = \frac{2\pi}{\lambda} n \cdot 2t \cos \theta$ where θ is the angle of incidence of the incident light beam. Therefore,

the peak wavelengths can be identified by the condition: $\frac{\delta}{2\pi} = m = \frac{n(\lambda) \cdot 2t \cos \theta}{\lambda_{peak}}$ where m must be an integer number and we have also included the wavelength-dependence of the refractive index, $n(\lambda)$.

From our measurements we actually have the series of $\lambda_{peak,exp}$ values. Therefore, we can see whether it is possible to find a thickness t that correctly predicts our experimental peak series. As can be observed in the figure on the left, when $t = 4228 \text{ nm} \approx 4.23 \mu\text{m}$ we can match very well the calculated Fabry-Perot transmission peaks (vertical dashed lines) with the peaks in the experimental transmission spectrum (red curve) or the dips in the experimental reflection spectrum (blue curve). According to the manufacturer, our GaN substrates have a thickness of $t_{nominal} = 4 \pm 1 \mu\text{m}$ and therefore the estimated thickness t is in excellent agreement with the expected one. Small deviations can be attributed to i) minor

variations between the used values of $n(\lambda)$, that we took from the literature, and the refractive index of our specific substrate and ii) the finite resolution of our experimental spectra (acquired with 2 nm resolution).

We have added a discussion of this model to the supplementary information S6.

3. Fig 2b shows a linear correspondence between power and sc current. What the authors expect at higher powers and thus higher lattice temperatures? Would the temperature impact on the scattering frequencies modify the relative number of collected ballistic electrons?

We thank the reviewer for this question. First-of-all we would like to point out that, according to a recent study (see Narang et al. J. Phys. Chem. C, 120, 2016), it is safe to assume that, under continuous wave illumination, the operation of hot carrier devices is linear. Entering a regime where multi-plasmon processes are relevant would instead require ultra-fast, intense laser-pulses, in order to avoid melting of the nanoantennas. Therefore, the linear correspondence between power and sc current is expected to hold in our device until, eventually, the nanoantennas re-structures and melting occurs. According to a recent study, performed by some of the co-authors (Ref. 34), we also know that for higher irradiation intensities a greater number of non-thermal carriers will be generated raising the temperature of the background thermal carriers faster. Concurrently, as the reviewer correctly suggests, higher temperatures would increase the electron-electron collision rate and hence reduce the carrier mean free path. Thus, it is to be expected that at high enough temperatures, a reduction of the fraction of the ballistic carriers would occur. However, as discussed in the next section, the localization of the electric field close to the metal/semiconductor interface has a favorable effect on carrier collection, enabling collection of a large fraction of the ballistic carriers that are generated close to the interface. Furthermore, particularly in our studied configuration, the large size and high thermal conductivity of the substrate limit the temperature rise of the plasmonic antennas to few tens of degrees even for MW/m^2 irradiation intensities. Therefore, the effect of temperature increase on hot carrier scattering is very limited for plasmonic hot carrier photodetection devices under continuous wave irradiation.

4. How the thickness of the stripes would influence the mix of ballistic vs scattered electrons collection?

We thank the reviewer for this important observation. Based on recent DFT calculations we know that, in Au, hot-electrons with 1.2 eV of excess energy (equal to our Schottky barrier height) have an expected mean free path of approximately 20 nm (versus the ~ 40 nm mean free path of electrons at the Fermi level) (Ref. 32). Hence, our initial design was chosen to enable ballistic collection of hot-electrons from within the entire volume of our nanoantennas. If the thickness of the antenna is increased, the contribution of scattered carriers is expected to increase, particularly for higher photon energies. In fact, the mean free path of hot-electrons decreases as the inverse of the excess energy squared (Ref. 32) and therefore only scattered electrons will propagate long-enough to reach the interface.

However, electron-electron scattering reduces the electron energy by a factor of 2 and, for low photon energies, this causes the scattered carrier to have insufficient energy for being collected above the Schottky barrier. Nonetheless, in the studied configuration, which is very common for plasmonic photodetectors, the plasmonic antenna sits on a high-index substrate (GaN) and therefore the electric field is localized within 10nm of the metal-semiconductor interface (see Figure 4a). Therefore, the non-uniform field profile inside the antenna mitigates the effect of increasing thickness and favors ballistic collection. For thicker stripes supporting higher order plasmon resonances, the situation would change abruptly. In fact, in the case of a quadrupole plasmon mode the field is localized also at the upper surface of the metal stripe, making the mix of ballistic versus scattered carriers much more sensitive to the overall thickness of the plasmonic antenna, scattered carrier collection being increasingly important for higher photon energies. We have added the following sentences in the revised manuscript to address this aspect:

“Increasing the thickness of the plasmonic antenna would increase the contribution of scattered carriers to the observed photocurrent, due to the increased distance that hot electrons must travel before reaching the interface. Since each electron-electron scattering event approximately reduces the electron energy by a factor of two, it is expected that scattered carriers would only provide a significant contribution at higher photon energies; those carriers created at lower photon energies would likely have insufficient energy to overcome the Schottky barrier. Nonetheless, in the studied configuration, which is very common for plasmonic photodetectors, the plasmonic antenna sits on a high-index GaN substrate and therefore, the electric field is localized close to the metal-semiconductor interface upon excitation of the fundamental plasmon mode (Figure 4a). The largest hot carrier generation thus occurs close to the interface, and as a result, the non-uniform field profile inside the antenna favors ballistic collection, mitigating the effect of increasing antenna thickness.”

5. What would happen to IQE shape (fig 3c for instance) by using a metallic material (e.g. aluminum) where the interband transition is at lower energy than the plasmonic resonance (for properly designed stripes)?

Theoretical calculations show that Al interband transitions produce a hot-electron/hot-hole distribution which is very similar to an intra-band distribution (see Ref. 32). Therefore the IQE spectrum would preserve the quadratic trend also for frequencies above the interband threshold of Al. We have added the following sentence in the revised manuscript to explicitly address how the metal band structure of a metal like Al would affect the shape of the IQE.

“Interestingly, recent DFT calculations³² show that in the case of Al nanoantennas, interband transitions produce a hot electron/hot hole distribution which is very similar to the intra-band case and therefore IQE could preserve the quadratic dependence on the photon-energy even above the inter-band threshold (~1.6 eV).”

6. Fig. 4c: the spike of the experimental (grey) at the right end of the plot is just an oscillation due to experimental instabilities or the IQE trend changes even further at higher energies?

The last point of the spectrum is acquired close to the upper limit of photon energies available from our supercontinuum laser. Therefore, in this range, the signal becomes noisy due to the low power of the laser. Hence, the observed spike is attributable to experimental instabilities of our detection scheme.

Reviewer #3 (Remarks to the Author):

Major comment: The authors studied about the role of surface plasmon excitation in hot carrier generation, transport and injection by exploring internal quantum efficiency of photoexcited Au-GaN Schottky diodes. The authors concluded that the light-harvesting capability and strong near-fields of surface plasmon do not alter carrier generation rates or the internal device physics governing carrier transport, and that plasmon excitation does not a priori selectively enhance the rate of any particular decay process or transport mechanism.

Q1. However, despite mentioning such important things, it is unclear why GaN which is not common for hot electron transfer studies was used.

Although the reasons for choosing GaN were described concisely in the manuscript, we will outline the rationale again here. In plasmonic hot carrier devices based on internal photoemission (IPE), the semiconducting substrate must satisfy three requirements: i) does not absorb any photons in the wavelength range of interest in order to prevent photoexcitation of free carriers within the semiconductor itself; ii) transports the collected hot carriers away from the interface with minimal electrical resistance; and iii) forms a Schottky barrier with the metal in order to favor the separation of the hot carriers and prevent their recombination. Indeed, n-type GaN satisfies all three requirements:

- i. Extremely wide bandgap (3.4eV) enabling the study of hot carrier processes across the near-IR, visible and UV spectrum;
- ii. Excellent electrical characteristics: due to its widespread use in the optoelectronic industry, n-type GaN can be purchased with various doping levels, with high electron mobility, including in the form of highly-doped low-resistance, crystalline substrates;
- iii. Appropriate conduction band edge, forming a Schottky barrier of ~ 1.2 eV with Au nanoantennas

Almost all previous reports of visible-frequency plasmonic hot carrier generated photocurrents have used TiO₂ as a semiconducting substrate, but have typically focused on characterizing the external quantum efficiency rather than the internal quantum efficiency. The reasons for this are not completely clear, but may be due to the compromise between electrical conductivity and optical transparency in TiO₂, which often requires post-growth doping to raise the conductivity to level needed for creation of optoelectronic devices (Zheng et al., Nat. Comm. 7797, 2015). Unfortunately, this approach can engender sub-bandgap absorption or scattering from defects in the TiO₂ support (n.b., the blue color of the device reported by Zheng et al.). This makes it difficult to discriminate visible-light absorption in TiO₂ from absorption in the Au in the back-illumination configuration used in our experiments, and scattering centers hinder the possibility of performing accurate transmission measurements, which are necessary to obtain the quantitative absorption of the Au nanoantennas required for IQE determination. By contrast epitaxial GaN maintains high conductivity and high transparency simultaneously.

In order to make the choice of GaN substrate obvious to the reader, we have modified a sentence in the manuscript:

“Accordingly, our experimental platform consists of planar Au plasmonic photodiodes on an optically-transparent yet highly electrically conductive n-type GaN substrate, that we have identified as an optimal support (see Methods) to enable coupled electrical and optical (both transmission and reflection) characterization throughout the entire ultraviolet/visible/near infrared (UV/Vis/NIR) spectral range”.

and we have elaborated upon our reasoning in the Method section:

“In order to perform coupled optical and electrical measurements of a plasmonic IPE device for experimental assessment of its IQE, it is necessary to have a semiconducting substrate which: i) does not absorb light in the wavelength range of interest in order to prevent interband photogeneration of carriers within the semiconductor and also does not scatter light (optically transparent); ii) has high electrical conductivity to enable transport of hot carriers; and iii) forms a Schottky barrier with the metal to favor separation of the hot electrons and holes and to

prevent their recombination. The n-GaN substrate employed here satisfies all of these requirements: i) it has a wide bandgap (3.4eV), and is optically transparent, with no light scattering centers; ii) due to its widespread use in optoelectronics, it is commercially available with various doping levels, in the form of highly-doped low-electrical resistance, crystalline substrates; iii) its band-alignment leads to the formation of a sizable Schottky barrier of ~1.2 eV with Au. These factors motivated our use of GaN as a semiconducting support for the study of plasmonic IPE devices.”

Q2. Especially, it is inappropriate to discuss in the same way as the past studies using Au-NPs loaded TiO2 showing EQE from 0.1 to 1% by using the devices with very low EQE and IQE in the order of 10^{-5} . IQE is the sum of all processes, and such discussion can only be done by measuring dynamics of each process such as dephasing of LSPR concerning with a hot carrier generation and excitation wavelength dependence of injection rate of hot electron into GaN electron conduction band. Furthermore, the near-field enhancement of Au line and space pattern is very low because the electromagnetic field is not localized strongly (Even its near-field spectrum has not been measured.). The authors are needed to perform experiments by using hot carrier devices with high EQE efficiency and Au nanostructures with high near-field enhancement including more detailed measurements of not only IQE but also near-field spectrum as well as the dynamics of each process. Otherwise, it cannot be believed that IQE was not dependent on the localized surface plasmon and its near-field enhancement. Therefore, this version of manuscript cannot be accepted for publication in Nature Communications.

We strongly disagree with the reviewer’s assertion that characterization of IQE for GaN is not useful, and we address here some of the crucial aspect of our work in relation to his/her comments. Our work focuses on in-depth understanding of carrier generation and transport in plasmonic hot carrier devices, and we assert that this mechanistic approach enables the identification of the fundamental bottlenecks limiting the overall quantum efficiency. We in fact design and characterize three plasmonic nanoantennas with identical material and interfacial properties and very similar plasmon peak energy (between 1.75 eV and 1.9 eV, showing the plasmon origin of antenna absorption via polarization-dependent absorption measurements). We point out here that our nanoantennas exhibit strong absorption (60% at the plasmon resonance), much larger than would be possible using just plasmonic nanoparticles (typically 10-15% depending on the size and surface coverage). As correctly noted by the reviewer, IQE is the convolution of several electronic processes. Yet, by design in these devices the electronic processes and their characteristic time- and length-scales are comparable at the plasmon resonance. Therefore, if plasmon excitation had any direct impact on the hot carrier transport and collection mechanisms, a plasmonic signature should appear in the IQE at the corresponding frequency. However, based on spectrally-resolved absorption and photocurrent measurements under steady state conditions (continuous wave illumination), this is not what we observe. We further note here that, contrary to the reviewer comments, the field enhancement in our system is comparable to that of several previous works on plasmonic hot carrier devices (see for example Zheng et al., Nat. Comm. 7797, 2015). Thus the absence of a plasmonic signature in the IQE is not attributable to differences in field enhancement (please note that Figure 4a reports the absolute magnitude of the electric field and not the field enhancement, we added the latter information to the revised Supplementary Information S7).

Instead, our experiments indicate that IQE specifically presents a signature of the electronic structure of the gold nanoantennas. As we discuss in the manuscript, in the intraband transport regime ($h\nu < 2\text{eV}$), which is where the plasmon resonances of all our devices lie, the IQE follows the conventional Fowler model for internal photoemission. At larger photon energies, the suppression of IQE can be related to the onset of interband transitions in gold. This result suggests that our hot carrier devices operate based on a three-step internal photoemission process and we thus conclude that, in this regime, plasmon excitation does not affect transport processes but solely influences the photon absorption. As we state in the discussion, we do not seem to observe any impact of interfacial plasmon excitation in these nanoantennas, whose characteristic dimensions are in the tens of nanometers. We note that interface plasmon generation has been recently reported for metallic nanocrystals and is characterized by a constant quantum efficiency as a function of the incident photon energy (Ref 44). We have stressed this statement in the revised manuscript to avoid any confusion of the reader concerning the involved processes:

“We also note here that plasmon-mediated interfacial hot carrier excitation has been observed in selected systems employing small metallic nanocrystals and constitutes a different mechanism for harnessing hot carriers beyond IPE^{44,45}. In fact, in the case of interfacial plasmon excitation the quantum efficiency has been shown to exhibit a stepwise efficiency spectrum with a system-specific threshold energy⁴⁴. However, in the studied systems, which have dimensions of several tens of nanometers, we can entirely ascribe the IQE spectral features to the metal properties and we do not observe any deviations that could be attributed to a competing contribution from interfacial plasmon excitation. “

We further note that our parameter-free model captures the details of the three-step photoemission process occurring in our devices and *quantitatively* agrees with our experimental results. This model, which has been recently validated also by experimental ultrafast spectroscopy data (Ref. 34), includes ab-initio calculated lifetimes of LSPR dephasing and generation of hot-carriers, as well as electron-electron and electron-phonon scattering processes. Therefore the dynamics suggested by the reviewer are already accounted for in our modelling. Considered that these dynamics have been extensively studied and reported numerous times over the past two decades (*J. Phys. Chem. B* **1999**, *103*, 8410; *Annu. Rev. Phys. Chem.* **2003**, *54*, 331; *J. Am. Chem. Soc.* **2007**, *129*, 14852.; *Chem. Rev.* **2011**, *111*, 3858.; *Nano Lett.* **2017**, *17*, 6047, Ref. 34) we believe that the measurement suggested by the reviewer are unnecessary to support our conclusions. Our model enables deeper insights into the operation of plasmonic hot carrier devices than have been previously possible. For example we can analyze the number of scattering events a carrier has undergone before being collected, information that cannot be determined with any experimental technique, to the best of our knowledge.

The simulated electric field is the starting point for our model, which is the reason why we prefer to report this metric in Figure 4a, and is obtained with widely accepted and validated electromagnetic simulation methods. Considering that our parameter-free model is in excellent agreement with our experimental result, we believe that near-field measurements would not provide new information about our devices and are hence unnecessary to support our conclusions. Importantly, we note that, as discussed in the manuscript, the maximum field strength occurs at the buried interface between the metal and the semiconductor. While near field techniques allow to map the field enhancement around the metallic nanoantennas, as reported many times in the literature, they don't enable direct measurement of the critical near field enhancement at this 'buried' interface. Instead, calculations offer the full picture.

Finally, we would like to point out that our devices have an estimated EQE of $\sim 0.001\%$ (10^{-5} is absolute fraction of unity, not in percentage as the reviewer seems to suggest), which is in line with previously reported devices in the literature (see *Knight et al. Science* **332**, *2011*). Based on our physical understanding of the device operation, the low EQE cannot be the reason why we do not observe a plasmonic signature in the IQE spectra. Importantly, thanks to our quantitative modelling, we were able to clearly detail the origin of the quantum efficiency losses in our devices. As discussed in the manuscript and SI, we observe that the Schottky barrier height and momentum matching conditions are critical to determining and enhancing the EQE and IQE of hot carrier devices based on three step photoemission process. In this respect, nanoparticle-based devices, mentioned by the reviewer, do offer some advantages as momentum matching conditions are known to relax at the interfaces in nanoscale structures. However, our goal in this manuscript is to reveal mechanisms using a well-defined model systems. Nanoparticles feature particle-size polydispersity, require more complicated electrical contacting schemes (e.g. requiring transparent conducting oxides) and offer less tunability of the optical properties than our larger scale model system plasmonic antennas.

[Redacted]

[Figure Redacted]

Minor comments:

Q4. This manuscript does not show the I-t curve responding to the light turning on and off and its photocurrent stability. I-V curve measurement is also important.

We thank the reviewer for these suggestions. We have added in the supplementary information S5, Figure S5a, the I-t trace we used to determine the power-photocurrent relation in our device (Figure 2b). Regarding the device stability, we note that it does not present any photodegradation as all materials (GaN and Au) are extremely stable under light irradiation. In the revised supporting information, Figure S5b we have also added the I-V curve of a hot carrier device.

Q5. By using polarization dependence (difference in EQE action spectra between transverse and longitudinal modes), the photocurrent contributed from interband transition can be eliminated in the case of the Au line and space patterns or Au nanorods with aligned orientation. Namely, EQE spectrum only derived from surface plasmon can be obtained. However, the authors did not show the polarization dependence.

We need to point out that the reviewer is incorrect about us showing polarization-dependent results. It was explicitly stated in the main manuscript text and shown in the supplementary information, that we have used polarized light for our experimental measurements and we have characterized the polarization dependence of both absorption (Figure S1) and photocurrent (Figure S2, dashed line in the responsivity graph). The addition of the transmission, reflection and absorption spectra for both light polarizations in Figure S2 shows that no absorption occurs at the plasmon resonance when the light is polarized parallel to the stripes. Hence, it is not necessary to eliminate the contribution of interband transitions from the photocurrent at the plasmon resonance because it entirely originates from intraband transitions. For polarization parallel to the stripes low absorption occurs for wavelengths shorter than 500 nm giving rise to a very low photocurrent signal due to the unfavorable hot carrier distribution.

Q6. The interference pattern was observed in the absorption spectrum because a GaN film with a thickness of about 4 micrometers is formed on a sapphire substrate. However, it is unclear why even the IQE spectrum also shows interference pattern. This indicates that the IQE spectrum was not obtained correctly.

We strongly disagree regarding the validity of our IQE measurements. Fabry-Perot interference is extremely sensitive to the light path within the high refractive index material (in our case GaN). The incomplete cancellation of the Fabry-Perot fringes is attributable to differences in the angular alignment of ± 3 degrees between the IQE and absorption measurements, which were done in separate apparatus. The plasmonic absorption is instead insensitive to angular variations of this magnitude. Therefore, the small discrepancy in Fabry-Perot fringes, which leads to their incomplete cancellation, has no bearing on the overall IQE measurement, nor it impacts the discussion related to the role of plasmon excitation, as also proven by the excellent agreement with our theoretical modelling.

Reviewers' comments:

Reviewer #1 (Remarks to the Author):

The authors have satisfactorily replied all my questions, as well as those from the other Reviewer. Therefore, I fully recommend this paper to be published.

Reviewer #2 (Remarks to the Author):

The authors satisfactorily addressed the comments and included the appropriate modifications in the main text and SI.

Therefore I can recommend the publication in Nature Communications.

Reviewer #3 (Remarks to the Author):

The authors emphasize in their reply that plasmon excitation does not affect transport processes but solely influences the light absorption because a plasmonic signature did not appear in the IQE at the corresponding frequency, so that they concluded that plasmon excitation had some direct impact on the hot carrier transport and collection mechanisms only by using GaN as n-type semiconductor photoelectrode. Importantly, this manuscript is written as the explanation can be applied to all the systems where plasmon-induced hot electron transfer from metallic nanostructure to n-type semiconductor takes place. Actually, in the previous article for elucidating IQE based on plasmon-induced hot electron transfer "Opt. Express 2017 doi.org/10.1364/OE.25.00A264", the experimental results and discussions are different from this study; IQE spectrum is affected by the near-field intensity, namely surface plasmon resonance. It means that the observed phenomenon differs in the different semiconductor substrates. Therefore, the reviewer asked whether not only Au stripes but also Au nanoparticles, and not only GaN but also TiO₂ and so on also lead to the same conclusion or not. The reviewer considers that the observed phenomenon in the present study can be only explained by the thermodynamically disadvantageous in the longer wavelength region based on the energy gap between the flat band potential of GaN and Fermi level of gold (Schottky barrier) although the excitation of surface plasmons contributes to IQE. Namely, it is considered that the difference in the Schottky barrier between Au/GaN (used in this study) and Au/TiO₂ (used in Opt. Express 2017 doi.org/10.1364/OE.25.00A264) is attributed to the difference of results between this manuscript and Opt. Express 2017. If the authors consider that the former study "Opt. Express 2017 doi.org/10.1364/OE.25.00A264" is wrong, the authors should verify whether the IQE spectrum does not depend on the near-field spectrum by using Au stripes fabricated on TiO₂.

The authors replied that it is not necessary to measure ultrafast dynamics because the phenomenon is already understood by the former work. However, the listed articles by authors are the dynamics of general plasmonic nanoparticles (e-e, e-ph scattering) such as J. Phys. Chem. B 1999, 103, 8410 (S. Link et al.), Annu. Rev. Phys. Chem. 2003, 54, 331 (S. Link et al.), and Chem. Rev. 2011, 111, 3858 (G. V. Hartland), and are Au/TiO₂ systems such as J. Am. Chem. Soc. 2007, 129, 14852 (A. Furube et al.) and Nano Lett. 2017, 17, 6047 (D. C. Ratchford et al.). Even using the Au/TiO₂ system in their papers, the electron transfer dynamics could not be well resolved due to the limited time-resolution of their apparatus. However, from the study "J. Am. Chem. Soc. 2007, 129, 14852", it can be understood that electron transfer is at least induced since the electrons injected into the conduction band of TiO₂ can be measured with infrared probe pulse. However, there is no evidence as for Au/nGaN system because the reported study is Au/TiO₂ system. Therefore, the reviewer asked the authors to measure the ultrafast dynamics. At

least, it is necessary to measure the transient absorption spectroscopy to evaluate how many electrons are transferred to the conduction band of nGaN at each pump wavelength as well as their electron transfer rates, and the authors should compare the electron injection efficiency as a function of pump wavelength with the measured IQE spectrum because it should depend on the obtained IQE spectrum.

As for the I-V curve, on the other hand, the diode property could be seen correctly. The reviewer recommends the authors to show the I-V curve also under light irradiation conditions at several incident wavelengths. The reviewer guess that the voltage at which the photocurrent is observed is only shifted from around 1.2 V to negative voltage.

As for the Fabry-Perot interference fringes in the IQE spectrum, the reviewer agrees with the authors' reply that it occurs due to differences in the angular alignment of ± 3 degrees between the EQE and absorption measurement. However, as the authors can also easily predict, the readers will certainly have questions why the IQE spectrum also shows the interference fringes if he or she knows the meaning of IQE. Therefore, the reviewer recommends the authors to add a brief explanation why such a fringe appear also in IQE spectrum to the revised manuscript for better presentation. By the way, the fringe of IQE spectrum obtained by Cu/nGaN system seems to be natural in the case that the authors showed the reviewers in the response-to-reviewer file.

Reviewer #3 (Remarks to the Author):

The authors emphasize in their reply that plasmon excitation does not affect transport processes but solely influences the light absorption because a plasmonic signature did not appear in the IQE at the corresponding frequency, so that they concluded that plasmon excitation had some direct impact on the hot carrier transport and collection mechanisms only by using GaN as n-type semiconductor photoelectrode. Importantly, this manuscript is written as the explanation can be applied to all the systems where plasmon-induced hot electron transfer from metallic nanostructure to n-type semiconductor takes place. Actually, in the previous article for elucidating IQE based on plasmon-induced hot electron transfer “Opt. Express 2017 doi.org/10.1364/OE.25.00A264”, the experimental results and discussions are different from this study; IQE spectrum is affected by the near-field intensity, namely surface plasmon resonance. It means that the observed phenomenon differs in the different semiconductor substrates. Therefore, the reviewer asked whether not only Au stripes but also Au nanoparticles, and not only GaN but also TiO₂ and so on also lead to the same conclusion or not. The reviewer considers that the observed phenomenon in the present study can be only explained by the thermodynamically disadvantageous in the longer wavelength region based on the energy gap between the flat band potential of GaN and Fermi level of gold (Schottky barrier) although the excitation of surface plasmons contributes to IQE. Namely, it is considered that the difference in the Schottky barrier between Au/GaN (used in this study) and Au/TiO₂ (used in Opt. Express 2017 doi.org/10.1364/OE.25.00A264) is attributed to the difference of results between this manuscript and Opt. Express 2017. If the authors consider that the former study “Opt. Express 2017 doi.org/10.1364/OE.25.00A264” is wrong, the authors should verify whether the IQE spectrum does not depend on the near-field spectrum by using Au stripes fabricated on TiO₂.

We thank the reviewer for clarifying his/her concerns. In the following, we discuss further the generality of our core result i.e. that the IQE of plasmon-enhanced internal photoemission (IPE) hot carrier devices is not affected by plasmon excitation.

In our manuscript, we tested three plasmonic Au/GaN devices with distinct plasmon resonances, and already showed that the IQE curves do not exhibit any feature that can be attributed to plasmon excitation (Figure 2g). This result agrees with previously reported calculations [Sundararaman et al., Nat. Comm. 5, 2014] predicting that, across the near-infrared-visible-UV spectrum, the energy and momentum distributions of hot carriers generated by plasmon decay are dominated by the band structure of the metal and not by the local electric-field direction. In Figure 3 of the manuscript, we also reported the energy distributions of hot electrons generated in Au for different photon energies (Fig. 3a,b), as obtained from *ab initio* calculations, and we discussed how the interplay with a Schottky barrier of ~1.2 eV determines the salient features of the experimentally observed IQE spectra (Fig. 3c).

Here we extend our analysis to different values of the Schottky barrier and find that:

- contributions from the metal electronic band structure always dominate the IQE spectra at high energies (>1.8 eV) and always predict a peak in the IQE curve, as observed experimentally.
- the interfacial Schottky barrier, acting as a high-pass energy filter, modulates the relative contribution of intra and interband transitions and defines the final shape of the observed IQE spectra

From these arguments it follows that the IQE spectrum of Au-based plasmon-enhanced IPE hot carrier devices is independent of the device geometry and that the specific semiconductor employed in different experiments contributes solely by determining the energy filtering properties of the interfacial Schottky barrier. To confirm our understanding we show that our model correctly predicts literature-reported

results for Au/TiO₂ IPE plasmonic hot carrier devices, in particular those cited by the reviewer, and that it correctly captures the differences between Cu- and Au-based devices.

Figure R1: **Interplay of metal band structure and Schottky barrier on the IQE.** a) IQE of an Au/semiconductor (electron effective mass of GaN $\sim 0.27 m_{0e}$) device as a function of the Schottky barrier height. The light-grey dashed lines represent the IQE predicted by the Fowler model. The dark-grey vertical dashed lines identify the four photon energies analyzed in part b; b) hot carrier energy distribution for four different photon energies (from top to bottom: 1.4 eV, 1.71 eV, 2.3 eV and 2.6 eV). Each column corresponds to a different Schottky barrier height (from left to right: 0.9 eV, 1.2 eV, 1.5 eV) and shows how the energetics of the interface filter the hot carriers energy distribution.

Figure R1.a shows the calculated IQE for an Au/semiconductor interface (electron effective mass of GaN $\sim 0.27 m_{0e}$ [Rheinlander, Neumann, Phys. Status Solidi (b) 64, 1974]) as a function of the interfacial

Schottky barrier height. Beyond the specificity of each curve we observe that some features are common to all the spectra:

- At low photon energies (close to the Schottky barrier) the IQE grows rapidly, roughly following the expected free electron-like Fowler behavior
- Around 1.7eV, the IQE growth is suppressed and the curve deviates significantly from the Fowler model
- Beyond ~ 2.3 -2.4 eV the IQE exhibits a peak and then a pronounced drop

It is straightforward that these features are directly associated with the Au electronic band structure:

- Up to ~ 1.6 eV, intraband transitions dominate hot carrier excitation. For this reason the IQE spectrum follows closely the Fowler model, which accounts only for free-electron energy distributions;
- Between 1.6 eV and 1.8 eV, interband transitions start to contribute significantly and the IQE deviates more prominently from the free-electron like Fowler model. The suppression of IQE can be understood by looking at Figure R1.b. Indeed, starting at around 1.7 eV, interband transitions induce a redistribution of the energy of the hot-electrons towards lower energies, close to the metal Fermi level. This reduces the number of high-energy electrons that can be collected with better efficiency than low-energy electrons and suppresses the IQE compared to expectations based solely on the Fowler model.
- Around ~ 2.3 -2.4 eV, interband transitions finally become the dominant mechanism for hot-electrons generation leading to a sharper drop in IQE, which then reaches a minimum around 2.6 eV.

The effect of the interplay between intra and interband transitions can be entirely explained by considering the metal electronic band structure. On the other hand, the energy filtering effect of the metal/semiconductor interface, which combines the energetics of the Schottky barrier and momentum matching conditions, modulates the relative prominence of these three different regimes (intraband regime up to 1.6 eV, mixed regime 1.7-2.3 eV, interband regime above 2.4 eV). In fact, we observe that:

- For a Schottky barrier of 0.9 eV, effective collection of high-energy electrons from intraband transitions leads to a peak in IQE around 1.7 eV, right before the onset of interband transitions. A slow decrease in IQE is observed in the mixed regime before a sharp drop-off occurs around 2.4 eV;
- For a Schottky barrier of 1.2 eV, the collection of hot electrons in the intraband regime is reduced and therefore the mixed regime between 1.7 eV and 2.3 eV still exhibits a slow growth of IQE. Indeed, the increased injection probability of the highest energy electrons compensates the overall reduced probability of intraband generated hot-electrons. Only the complete transition to the interband regime around 2.4 eV suppresses the IQE;
- For an even larger Schottky barrier of 1.5 eV, carrier collection is negligible across the entire intraband regime. Therefore the transition between the Fowler-like rapid growth in IQE and the IQE reduction due to interband transitions cannot be observed. Instead, throughout the mixed regime, IQE slowly grows thanks to the highly energetic fraction of hot-electrons generated through intraband transitions. Transition into the interband regime around 2.4 eV, however, halts this growth.

From this analysis we observe that due to the combined effect of the metal band structure and the interface, the IQE spectrum can exhibit a peak in the photon energy range anywhere between ~ 1.8 eV to 2.4 eV. Although small gold nanoparticles or thin Au films can exhibit a plasmon resonance in this very same energy range, these two aspects should not be confounded. Indeed, plasmon excitation will affect the optical properties of the system and manifest itself in an enhanced external quantum efficiency of the device. However the peak in IQE originates solely from the electronic properties of the metal, with modulations induced by the characteristics of the metal/semiconductor interface. Measurements on multiple devices with distinct plasmon energies, like those we performed in this manuscript, in fact prove that the features of IQE are insensitive to plasmon excitation. Also, our results demonstrate that, by accounting for the metal and interface properties, in particular the Schottky barrier height, it is possible to predict all the characteristics of the experimental IQE spectra.

As a further proof of our understanding, we apply the same modelling used for the Au/GaN results reported in the manuscript to two closely related systems:

- a) Au/TiO₂ system (data from literature) – based on our understanding that the metal band-structure dominates the device IQE, we expect that this system will present similar features to the Au/GaN case. Yet, small variations can be expected due to the difference in Schottky barrier height and TiO₂ effective electron mass.

[Redacted]

In the cited Opt. Expr. work, the authors study an Au/TiO₂ device and observe a non-monotonic IQE spectrum. In particular, they attribute the peak in IQE around 2.2 eV to plasmon excitation and propose a model based on plasmon-induced anisotropy in the hot carrier momentum.

Figure R2 shows the IQE predicted using our ab initio electronic structure calculations compared to the experimental data reported in the cited Opt. Expr. paper. The only two input values we used are the reported 1.53 eV Schottky barrier height for the Au/TiO₂ interface and the effective electron mass of TiO₂ ($\sim 30 m_{e0}$ [Enright, Fitzmaurice, J. Phys. Chem 1996]). This model, which takes into account the precise details of the metal band structure and the energetics of the interface, clearly captures very well the experimental data without the need to invoke an anisotropic momentum distribution of the hot carriers due to plasmon excitation. The small offset in energy between the calculated values and the measured values can be attributed to the $\sim 10\%$ accuracy in the DFT calculations of the position of the d-bands of Au.

Figure R2: IQE of an Au/TiO₂ hot carrier device. Blue squares: data from Opt. Expr. Red Dashed line: our model calculation.

We further note that the free-electron model for plasmon-induced hot carrier anisotropy proposed in the Opt. Expr. paper is fundamentally inappropriate, since it assumes that constant energy surfaces remain spherical even at energies 1.8-3 eV above and below the Fermi level of Au. However, it has been shown [Sundararaman et al., Nat. Comm. 5, 2014] that at these energies for Au, the momentum distribution is dominated by the band structure and not by the electric field directions. In other words, the fact that the Drude model is still justified for computing the dielectric function of the metal at these energies does not imply that the involved states are free-electron like. Consequently, it is incorrect to assume that the momentum of hot carriers in these states can be directly determined by the plasmon electric field, ignoring the band structure. With our model, instead, we correctly capture the energy distribution of the hot-electrons and we average their momentum distribution over different crystal directions in order to account for the poly-crystallinity of our metal structures.

[Redacted]

[Figure Redacted]

[Redacted]

We therefore conclude by reaffirming the generality of the observation reported in the manuscript, i.e. that plasmon excitation does NOT affect the IQE of plasmon-enhanced IPE devices. In other words, as we show in our work using three Au/GaN devices with distinct plasmon resonances, plasmon excitation only contributes to absorption without modifying the mechanisms leading to the generation of hot carriers, or the characteristics of the generated hot carriers, i.e. plasmon excitation does not alter the hot carriers momentum. Instead, as we discuss in our manuscript, and have more extensively detailed in this response, the spectral shape of the IQE is controlled primarily by the metal band-structure. Additionally, the energetics of the metal/semiconductor interface (Schottky barrier height and momentum matching conditions) modulate the characteristic features determined by the metal band structure.

In this response we also showed that our model can be successfully applied to reported literature data [Redacted]. The excellent agreement between our modelling and experiments for a wide variety of device designs (3x Au/GaN devices, [Redacted] and 1x Au/TiO₂ device) provides solid evidence for our understanding of the role of plasmon excitation in hot carrier devices.

In order to help readers appreciate the secondary role of the Schottky barrier height, we have now added Figure R1 and the associated discussion to the Supplementary Information.

The authors replied that it is not necessary to measure ultrafast dynamics because the phenomenon is already understood by the former work. However, the listed articles by authors are the dynamics of general plasmonic nanoparticles (e-e, e-ph scattering) such as J. Phys. Chem. B 1999, 103, 8410 (S. Link et al.), Annu. Rev. Phys. Chem. 2003, 54, 331 (S. Link et al.), and Chem. Rev. 2011, 111, 3858 (G. V. Hartland), and are Au/TiO₂ systems such as J. Am. Chem. Soc. 2007, 129, 14852 (A. Furube et al.) and Nano Lett. 2017, 17, 6047 (D. C. Ratchford et al.). Even using the Au/TiO₂ system in their papers, the electron transfer dynamics could not be well resolved due to the limited time-resolution of their apparatus. However, from the study “J. Am. Chem. Soc. 2007, 129, 14852”, it can be understood that electron transfer is at least induced since the electrons injected into the conduction band of TiO₂ can be measured with infrared probe pulse. However, there is no evidence as for Au/nGaN system because the reported study is Au/TiO₂ system. Therefore, the reviewer asked the authors to measure the ultrafast dynamics. At least, it is necessary to measure the transient absorption spectroscopy to evaluate how many electrons are transferred to the conduction band of nGaN at each pump wavelength as well as their electron transfer rates, and the authors should compare the electron injection efficiency as a function of pump wavelength with the measured IQE spectrum because it should depend on the obtained IQE spectrum.

As for the I-V curve, on the other hand, the diode property could be seen correctly. The reviewer recommends the authors to show the I-V curve also under light irradiation conditions at several incident wavelengths. The reviewer guess that the voltage at which the photocurrent is observed is only shifted from around 1.2 V to negative voltage.

The diode behavior of the device is already shown in the supporting information, Figure S5, from which the Schottky barrier height of 1.2 eV was obtained. We have also shown the photocurrent response of the Au/n-GaN device under visible-light illumination at the peak plasmon wavelength of 650 nm at various powers in the supporting information (Figure S5). We have therefore established the diode behavior of the device and from it we have derived the crucial properties of the metal/semiconductor interface. In addition, the external quantum efficiency (EQE) of the device shown in Figure 2 of the main manuscript shows the wavelength-dependent photocurrent response of the device under light irradiation. It is therefore unclear how these additional measurements, would help clarify the main conclusions of our manuscript regarding the influence of metal band structure and Schottky-barrier height on the IQE of plasmonic hot carrier devices.

As for the Fabry-Perot interference fringes in the IQE spectrum, the reviewer agrees with the authors' reply that it occurs due to differences in the angular alignment of ± 3 degrees between the EQE and absorption measurement. However, as the authors can also easily predict, the readers will certainly have questions why the IQE spectrum also shows the interference fringes if he or she knows the meaning of IQE. Therefore, the reviewer recommends the authors to add a brief explanation why such a fringe appear also in IQE spectrum to the revised manuscript for better presentation.

[Redacted]

We thank the reviewer for this suggestion. We have added a sentence to the manuscript referring to this discussion which is now provided in Supplementary Information S7.

REVIEWERS' COMMENTS:

Reviewer #3 (Remarks to the Author):

The authors have successfully reproduced the IQE spectrum by their numerical model even using the IQE data of Au/TiO₂ system reported in Opt. Express which the reviewer raised in the previous review process and **[Redacted]**. These results and discussions are convincing. However, the authors still do not consider about dynamics such as plasmon dephasing and hot electron transfer dynamics which the reviewer pointed out previously. The authors only performed the steady-state photoelectric measurement as an experimental approach for this study. Therefore, it is pointed out and recommended to describe in the main text that it was not performed based on the approach of ultrafast dynamics.

Reviewer #3 (Remarks to the Author):

The authors have successfully reproduced the IQE spectrum by their numerical model even using the IQE data of Au/TiO₂ system reported in Opt. Express which the reviewer raised in the previous review process and [Redacted]. These results and discussions are convincing. However, the authors still do not consider about dynamics such as plasmon dephasing and hot electron transfer dynamics which the reviewer pointed out previously. The authors only performed the steady-state photoelectric measurement as an experimental approach for this study. Therefore, it is pointed out and recommended to describe in the main text that it was not performed based on the approach of ultrafast dynamics.

We thank the reviewer for his comment. Following his suggestion we have now explicitly stated in the main text that EQE and Absorption, from which we derive IQE, are measured under steady state conditions. Furthermore, in the conclusions we explicitly point towards the necessity of elucidating further these processes using ultra-fast spectroscopy techniques.